# Early-Life Exposure to Environmental Air Pollution and Autism Spectrum Disorder: A Review of Available Evidence

**DOI:** 10.3390/ijerph18031204

**Published:** 2021-01-29

**Authors:** Giovanni Imbriani, Alessandra Panico, Tiziana Grassi, Adele Idolo, Francesca Serio, Francesco Bagordo, Giovanni De Filippis, Donato De Giorgi, Gianfranco Antonucci, Prisco Piscitelli, Manuela Colangelo, Luigi Peccarisi, Maria Rosaria Tumolo, Roberto De Masi, Alessandro Miani, Antonella De Donno

**Affiliations:** 1Department of Biological and Environmental Sciences and Technology, University of Salento, via Monteroni 165, 73100 Lecce, Italy; giovanni.imbriani@unisalento.it (G.I.); alessandra.panico@unisalento.it (A.P.); adele.idolo@unisalento.it (A.I.); francesca.serio@unisalento.it (F.S.); francesco.bagordo@unisalento.it (F.B.); antonella.dedonno@unisalento.it (A.D.D.); 2Local Health Authority ASL Le, 73100 Lecce, Italy; giov.defilippis@gmail.com (G.D.F.); donatodg@libero.it (D.D.G.); antongian2011@libero.it (G.A.); priscofreedom@hotmail.com (P.P.); ginopeccarisi@libero.it (L.P.); dmsrrt@gmail.com (R.D.M.); 3Medical Professional Association (OMCEO), 73100 Lecce, Italy; 4Italian Association of Health, Environment and Society (AISAS), via De Gasperi 22, Lizzanello, 73023 Lecce, Italy; manuela.colangelo95@gmail.com; 5Research Unit of Brindisi, c/o ex Osp. Di Summa, Institute for Research on Population and Social Policies, National Research Council, Piazza Di Summa, 72100 Brindisi, Italy; mariarosaria.tumolo@irpps.cnr.it; 6c/o Campus Ecotekne via Monteroni, Branch of Lecce, Institute of Clinical Physiology, National Research Council, 73100 Lecce, Italy; 7Multiple Sclerosis Centre, Laboratory of Neuroproteomics, “Francesco Ferrari” Hospital, 73042 Casarano, Italy; 8Italian Society of Environmental Medicine, 02100 Milan, Italy; alessandro.miani@gmail.com; 9Department of Environmental Science and Policy, University of Milan, 02100 Milan, Italy

**Keywords:** autism spectrum disorder, environmental air pollution, early life exposure, pregnancy

## Abstract

The number of children diagnosed with Autism Spectrum Disorder (ASD) has rapidly increased globally. Genetic and environmental factors both contribute to the development of ASD. Several studies showed linkage between prenatal, early postnatal air pollution exposure and the risk of developing ASD. We reviewed the available literature concerning the relationship between early-life exposure to air pollutants and ASD onset in childhood. We searched on Medline and Scopus for cohort or case-control studies published in English from 1977 to 2020. A total of 20 articles were selected for the review. We found a strong association between maternal exposure to particulate matter (PM) during pregnancy or in the first years of the children’s life and the risk of the ASD. This association was found to be stronger with PM_2.5_ and less evident with the other pollutants. Current evidence suggest that pregnancy is the period in which exposure to environmental pollutants seems to be most impactful concerning the onset of ASD in children. Air pollution should be considered among the emerging risk factors for ASD. Further epidemiological and toxicological studies should address molecular pathways involved in the development of ASD and determine specific cause–effect associations.

## 1. Introduction

Autism Spectrum Disorder (ASD) can be defined as a set of heterogeneous conditions of neurological development characterized by early difficulties in social communication and unusual repetitive behaviors along with limited interests, some of which can be attributed to several etiological factors including Mendelian mutations of individual genes or resulting from complex interactions between genetic and non-genetic risk factors [1].

The most recent American data from the US Centers for Disease Control and Prevention (CDC), regarding year 2016, report a prevalence of ASD up to 1 child out of 54 in pediatric population aged 8 years old [2]. The overall prevalence of ASD has grown exponentially since 2000, passing from 1 case out of 150 new births (in children born in 1992) to 1 case out of 59 in 2014, concerning children born in 2006 [3,4].

According to the World Health Organization (WHO), 1 in 160 children suffer from ASD [5]. This estimate represents an average figure obtained by analyzing several studies produced globally. Epidemiological studies carried out over the past 50 years seem to show a constant increase in prevalence rates. There are many possible explanations for this rapid increase, including better awareness, expansion of diagnostic criteria [6], better diagnostic tools, and better reporting [7,8]; but all these conditions cannot explain the huge expansion of the overall number of ASD subjects.

At the basis of its complex etiology, there is a genetic component that has been confirmed over the years by the identification of mutations of several high-confidence genes involved in the neuronal and cortical organization, in the formation and maturation of synapses, and in neurotransmission [9,10,11,12], that can lead to a transient interruption in the brain’s development process [13].

In an epigenetic perspective, environmental factors have recently emerged as extremely important contributors to the etiology and pathophysiology of ASD, so that a better understanding of these gene–environment interactions is now fundamental [14].

Air pollution is one of the most studied environmental factors which seems to play an important role in the etiology of ASD. Several studies have identified in recent years a positive association between exposure to different environmental pollutants—such as atmospheric particulate matter (PM_2.5_ and PM_10_), nitrogen dioxide (NO_2_), sulfur dioxide (SO_2_), carbon monoxide (CO), and ozone (O_3_) [15]—and the onset of neurodevelopmental diseases, including ASD [16]. The neurotoxicity mechanisms of these pollutants have not been fully understood. However, inflammation and oxidative stress processes seem to play a crucial role in the structural and functional changes affecting the central nervous system (CNS) that may cause mental disorders [17,18], possibly mediated by epigenetics modifications of the genes involved in these processes [19].

Another key concept for understanding causation of ASD concerns the “critical time window”, as symptoms commonly appear in the early years of the baby’s life, thus suggesting a strong probability of origin during the preconceptional, pregnancy, and early stages of life [14]. Studies on human pregnancy and in the early years of life found that prenatal exposure to air pollution during the development and maturation phase of the CNS can be linked to a systemic inflammatory response, which consequently causes an impairment in the neurodevelopment process [20]. Therefore, we aimed at carrying out a comprehensive review of potential associations between regularly monitored air pollutants and ASD, and, at same time, analyzing the potential factors that could influence this relationship (i.e., “exposure window” and the different covariates that are considered in scientific literature). The results obtained on the basis of the scientific evidence reported in the literature could provide more detailed and reliable information on the possible interaction between air pollutants and ASD, which could help epidemiologists and other researchers to carry out further studies to investigate this potential association.

## 2. Materials and Methods

We reviewed the available literature concerning the relationship between early-life exposure to air pollutants and ASD onset in childhood. The potential factors that could influence this relationship, such as “exposure window” and the different covariates are displayed in scientific literature. The review was carried out according to the main items reported in the PRISMA checklist 2009 [21]. We searched on Medline and Scopus for cohort or case-control studies published from 1977 to 2020, by using the following keywords: “autism spectrum disorder” AND “air pollution” OR “particulate matter” OR “environmental pollutants” OR “polycyclic aromatic hydrocarbons” OR “nitrogen dioxide” OR “PM2.5” OR “PM10” OR “traffic-related pollutants” AND “prenatal” OR “pregnancy” OR “early life”.

The bibliographic search was carried out separately in each database by selecting the studies based on their titles or abstracts and subsequently, the entire articles were analyzed. Studies were considered for the review according to specific inclusion criteria:human studies;full-text articles in English;assessment of exposure to airborne environmental pollutants during pregnancy or in the first years of the baby’s life;only case-control studies or cohort studies.

For each examined study, we extrapolated the following information:authors and year;study design;sample size;exposure assessment measures;method of analysis;adjustment variables;ASD outcome assessment;time window of exposure;main findings.

A total of 20 articles were selected for the review. Exploration of heterogeneity of the studies was performed by assessing their quality (i.e., level of evidence). Interpretation of the findings was conducted in the frame of current knowledge [22].

## 3. Results

We retrieved a total of 575 references from our literature search, of which 54 met inclusion criteria based on title and abstract screening and 20 further met inclusion criteria based on full text screening (Figure 1).

### 3.1. Association between PM and ASD

We found 12 studies (9 case-control and 3 cohort studies) investigating the association between the risk of ASD onset and maternal exposure to PM (PM_2.5_ and PM_10_) during pregnancy and in the first years of the child’s life and the risk ASD onset (Table 1). The case-control studies showed a positive association between PM exposure (especially PM_2.5_) during pregnancy or early life and ASD [23,24,25,26,27,28,29,30,31], while the cohort studies demonstrated a less significant relationship.

A recent study involving 674 children with ASD and 855 controls recruited in the Study to Explore Early Development carried out in the United States found a positive association between exposure to PM_2.5_ in children’s first year of life and ASD. The exposure to PM_2.5_ in the different periods ranging from three months before pregnancy, the three trimesters of pregnancy and the first year of life, was assessed using a satellite-based model. The strongest association was found between PM_2.5_ in the first year of life, with an odds ratio (OR) of 1.3 (95%CI: 1.0, 1.6) for 1.6 µg/m^3^ increase in PM_2.5_ [23].

The same population was analyzed in another study carried out in 2019 to assess the effects of the exposure to PM_2.5_ generated nearby roadways during the three trimesters of pregnancy and during the first year of life of the children. In this study, the deprivation index was also assessed as indicator of socio-economic status of the examined population. The association between exposure to PM_2.5_ and the risk of ASD in the first year of life was found stronger among those presenting a higher deprivation state (OR = 2.42, 95%CI = 1.20, 4.86) than those living in a low deprivation state (OR = 1.46, 95%CI = 0.80, 2.65) [24].

A recent study published in 2019 assessed the association between ASD and PM_2.5_ during the three trimesters of pregnancy and in the first two years of life, identifying a correlation between ASD and the increase in PM_2.5_ concentrations. The study was conducted on a group of 428 cases and 6420 controls in Ohio and used the daily PM_2.5_ individual exposure estimations provided by the United States Environmental Protection Agency (US EPA), based on their residential address. The analysis showed that the ORs related to second trimester, the first year of life and the cumulative period (that included the period from pregnancy to the second year of life) were: 1.41 (95%CI: 0.89, 2.24), 1.54 (95%CI: 0.98, 2.40), 1.52 (95%CI: 1.00, 2.31), respectively [25].

A study conducted on the Danish population that enrolled 15,387 children with ASD and 68,139 controls, examined exposure to various environmental pollutants, during the 9 months of pregnancy and in the following 9 months of life using estimates provided by the dispersion models (AirGIS) for the detected pollutants (NO_2_, SO_2_, PM_2.5_, and PM_10_) in relation to the maternal residence during the examined periods. The data obtained showed that exposure to PM_2.5_ and PM_10_ during the 9 months after pregnancy increased the risk of a diagnosis of ASD; in particular, the adjusted ORs for PM_2.5_ was of 1.06 (95%CI: 1.01, 1.11) per interquartile range (IQR) increased by 3.61 μg/m^3^, while an OR of 1.04 (95%CI: 1.01, 1.15) per each increase of 3.8 μg/m^3^ in IQR value was observed for PM_10_ [26].

A case-control study developed in Shanghai that enrolled 124 cases and 1240 controls assessed the relationship between exposure to PM_2.5_ and PM_10_ during the first three years of life and the risk of developing ASD. Using satellite data to evaluate the median levels of PM and performing a logistic regression model to evaluate the association between PM_2.5_, PM_10_ and ASD, the researchers observed that there was a strong association. In particular, for PM_2.5_ they calculated an OR of 1.50 (95%CI: 1.01, 2.22) in the second year of life and an OR of 1.78 (95%CI: 1.05, 3.02) in the third year, respectively for each increase of 3.4 μg/m^3^ in the IQR. As far as PM_10_, a clear association with ASD during the first three years of life was also found [27].

A 2015 case-control study involving 245 ASD cases and 1522 controls recruited from the Nurses’ Health Study II (NHS II) study—whose mothers lived at the same address before and after the pregnancy—found that higher exposure to PM_2.5_ during pregnancy and, in particular, during the third trimester was associated with an increased risk of subsequent ASD diagnosis. Exposure to PM_2.5_, assessed by using a space–time model and linked to maternal residence addresses, showed an OR of 1.42 (95%CI: 1.09, 1.86) per each increase of 4.40 μg/m^3^ in IQR value during the third quarter. This association was weaker during the first two quarters (ORs = 1.06 and 1.00). On the contrary, no significant association was found for PM_10_ [28].

Another study involving 217 cases and 226 controls in southwestern Pennsylvania evaluated the association between prenatal and early life exposure to PM_2.5_ and the risk of ASD. The results showed that both prenatal exposures and those occurred in the first two years of life were associated with an increased risk of ASD, expressed by an OR of 1.51 (95%CI: 1.10, 2.26) [29].

In 2013, a study involving 7603 children diagnosed with ASD, born in Los Angeles (California) from 1998 to 2009, and a high sample of paired controls with a ratio of 1:10, specifically assessed the influence of exposure to environmental pollutants from vehicular traffic during pregnancy on the onset of ASD. The study was performed by using a Land Use Regression (LUR) model, based on predictable pollution models, to estimate the concentrations of pollutants in a given area and correlating maternal residences to the environmental monitoring stations closest to them. The results highlighted clear associations between the onset of ASD and prenatal exposures to PM_2.5_ [30]. In the same year, additional analyses performed on the same population confirmed that exposure to PM_2.5_ during pregnancy and in the first year of life was associated with ASD. Moreover, an association between exposure to PM_10_ in the prenatal period and the risk of ASD was also found [31].

A 2019 retrospective cohort study that involved 246,420 children born in Kaiser Permanente Southern California (KPSC) from 1999 to 2009, assessed the risk of developing ASD associated with prenatal and first-year of life exposures to PM_2.5_. The results revealed a statistically significant association between the exposures to PM_2.5_ and the risk of ASD quantified by the following Hazard Ratios (HR) per each increase of 6.5 μg/m^3^ in IQR values in different phases of the pregnancy: 1.17 (95%CI: 1.04, 1.33) in the entire nine-months period; 1.10 (95%CI: 1.02, 1.19) in the first trimester of pregnancy, 1.08 (95%CI: 1.00, 1.18) in the third trimester of pregnancy; 1.21 (95%CI: 1.05, 1.40) in the first year of life [32].

At the opposite, a 2019 cohort study involving children born in Metro Vancouver, British Columbia, and Canada, born from 2004 to 2009, found no significant association between PM_2.5_ and ASD [33].

Another 2016 cohort study involving cohorts from four big European studies, CATSS (Sweden), Generation R (the Netherlands), GASPII (Italy), and INMA (Spain), assessed maternal exposure to PM_2.5_ and PM_10_ in the different periods of pregnancy and the risk of ASD without identifying statistically significant associations [34].

A 2015 case-control study conducted on 979 children with ASD and 14,666 controls —recruited between North Carolina and San Francisco Bay in California—assessed maternal exposure to PM_10_ during the three trimesters of pregnancy by interpolating (geostatistical interpolation) the daily concentrations of PM_10_ with a geostatistical method which considered the residence addresses of the mothers at the time of birth. The results obtained showed an association with the third trimester of pregnancy, with an OR of 1.36 (95%CI: 1.13, 1.63) per 10 μg/m^3^ increase in IQR value [35].

We also found some case-control studies which documented no significant statistical association between PM_10_ exposures and ASD onset. A 2018 study performed in Tehran involving 134 children with ASD and 388 controls investigated the association between long-term exposure to environmental pollutants and an increased risk of ASD diagnoses in children aged 2–10 years old. The results showed no association between the average exposure levels of mothers during pregnancy to PM_10_ and the onset of ASD [36].

A European study carried out on 5136 children with ASD and 18,237 controls living in Stockholm (Sweden) evaluated PM_10_ levels resulting from vehicular traffic by using dispersion models in relation to the addresses of maternal residence during pregnancy or the first year of the child. The results showed no association between exposures to PM_10_ and the risk of ASD [37].

Another study conducted on the Swedish population did not find clear associations between PM_10_ exposures during pregnancy, the first year of life, and the first nine years of life with ASD [38].

In the context of cohort studies, another study conducted in 2013 in Taiwan on a population of 49,073 children did not provide evidence of an association between PM_10_ in the first four years of life and an increased risk of ASD diagnosis [39].

Results for association between PM and ASD are shown in Table 1.
ijerph-18-01204-t001_Table 1Table 1Studies investigating the association between exposure to particulate matter (PM) and Autism Spectrum Disorder (ASD) development.Authors and YearStudy DesignSample Size/CountryASD Outcome AssessmentExposure AssessmentMethod of AnalysisAdjustment VariablesTime Window of ExposureMain FindingsMcGuinn et al., 2020 [23]Case-control674 ASD cases and 855 controls from California, Colorado, Georgia, Maryland, North Carolina, and PennsylvaniaASD case classification based on the results from the ADOS (Autism Diagnostic Observation Schedule) and ADI-R (Autism Diagnostic Interview-Revised)Satellite-based model to assign air pollutant exposure (PM_2.5_, O_3_) averages during several critical periods of neurodevelopmentLogistic regression model-study site -maternal age -maternal education -maternal race -maternal smoking, -month and year of birth.-3 months before pregnancy -each trimester of pregnancy -the entire pregnancy -first year of lifeThe strongest association was found between PM_2.5_ in the first year of life, with an odds ratio (OR) = 1.3 (95%CI: 1.0, 1.6) for 1.6 µg/m^3^ increase in PM_2.5_Kaufman et al., 2019 [25]Case-control428 ASD cases and 6420 controls from metropolitan Cincinnati area in southwest OhioASD diagnosis according to International Classification of Diseases, Ninth Revision, Clinical Modification (ICD-9-CM).Daily PM_2.5_ individual exposure estimations provided by the United States Environmental Protection Agency (US EPA), based on their residential addressLogistic regression model-maternal and birth-related confounders -multiple temporal exposure windows-each trimester pregnancy -first year of life -second year of lifeOdds Ratios related to second trimester, the first year of life and the cumulative period (that included the period from pregnancy to the second year of life) were: 1.41 (95%CI: 0.89, 2.24), 1.54 (95%CI: 0.98, 2.40), 1.52 (95%CI: 1.00, 2.31), respectivelyMcGuinn et al., 2019 [24]Case-control674 ASD cases and 855 controls from California, Colorado, Georgia, Maryland, North Carolina, and PennsylvaniaASD case classification based on the results from the ADOS (Autism Diagnostic Observation Schedule) and ADI-R (Autism Diagnostic Interview-Revised)Air pollution was assessed by roadway proximity and particulate matter <2.5 µm (PM_2.5_) exposureLogistic regression model-neighborhood deprivation-pregnancy -first year of lifeThe association between exposure to PM_2.5_ and the risk of ASD in the first year of life was found stronger among those presenting a higher deprivation state (OR = 2.42, 95%CI = 1.20, 4.86) than those living in a low deprivation state (OR = 1.46, 95%CI = 0.80, 2.65)Ritz et al., 2018 [26]Case-control15,387 ASD cases and 68,139 controls from DenmarkASD diagnosis according to International Classification of Diseases, Tenth Revision, Clinical Modification (ICD-10-CM)Exposures estimates provided by the dispersion models (AirGIS) for the detected pollutants (NO_2_, SO_2_, PM_2.5_ and PM_10_) in relation to the maternal residence during the examined periodsConditional logistic regression-parental age -neighborhood -socio-economic indicators -maternal smoking-pregnancy -9 months after pregnancyExposure to PM_2.5_ and PM_10_ during the 9 months after pregnancy increases the risk of a diagnosis of ASD with an adjusted OR of 1.06 (95%CI: 1.01, 1.11) for 3.61 μg/m^3^ increase of PM_2.5_
and an OR of 1.04 (95 % CI: 1.01, 1.15) per each increase of 3.8 μg/m^3^ PM_10_ in IQR value, respectivelyChen et al., 2018 [27]Case-control124 ASD cases and 1240 controls from ChinaASD cases were diagnosed by pediatricians according to Diagnostic and Statistical Manual of Mental Disorders, 5th Edition (DSM-V)Air pollution was assessed with satellite remote sensing dataConditional logistic regression-birth weight -gestational weeks -disease history -trauma history -maternal age, -familial mental health history -parents’ marital status -parental relationship -parenting, -parents’ educational level -smoking status-first year of life -second year of life -third year of lifeStrong association with PM_2.5_ in the second and third years of life with an OR of 1.50 (1.01, 2.22) in the second year and an OR 1.78 (1.05, 3.02) in the third year, respectively for each increase of 3.4 μg/m^3^ in the IQR Strong association with PM_10_ during the first three years of life and ASD with an OR of 1.16 (0.91, 1.49) in the first year, 1.73 (1.11, 2.68) in the second year and 1.58 (0.98, 2.56), per interquartile range (IQR) increase per 4.9 μg/m^3^Raz et al., 2015 [28]Case-control245 ASD cases and 1522 controls from United StatesFinal ASD case classification was based on the results from the ADI-R Autism Diagnostic Interview-Revised) and SRS scores (Social Responsiveness Scale)Exposure to PM_2.5_ and PM_10_ assessed by using a space-time model and linked to maternal residence addressesLogistic regression model-child’s birth year -birth month -sex -maternal age at birth -paternal age at birth-9 months before pregnancy -trimester 1,2,3 -entire pregnancy -9 months after birthAssociation with PM_2.5_ during the third trimester of pregnancy and ASD. OR of 1.42 (1.09, 1.86) per IQR increase per 4.40 μg/m^3^
No significant association between PM_10_ and ASDTalbot et al., 2015 [29]Case-control217 ASD cases and 226 controls from southwestern PennsylvaniaA case of ASD was defined as any child who scored a 15 or above on the Social Communication Questionnaire (SCQ), and had written documentation, including the Autism Diagnostic Observation Schedule (ADOS) or diagnosis of an ASD from a child psychologist or psychiatristPerson- and time specific PM_2.5_ estimates for individual and cumulative key developmental time periodsLogistic regression model-college education -smoking status -race -maternal age-pre-pregnancy -each trimester, pregnancy -first year of life -second year of life -cumulative (starting from pre-pregnancy)Both prenatal exposures and those occurred in the first two years of life are associated with an increased risk of ASD expressed by an Odds Ratio of 1.51, 95%CI = 1.10–2.26)Becerra et al., 2013 [30]Case-control7603 ASD cases and 10 controls per case from Los Angeles, CaliforniaThe diagnosis of ASD was based on the Diagnostic and Statistical Manual of Mental Disorders, 4th Edition, Text Revision (DSM-IV-R)Modeled concentrations of PM_2.5_ from air monitoring station 1993-2006, assigned by residential address at delivery/birthConditional logistic regression-maternal age -maternal place of birth -race -education -type of birth -parity -insurance type -gestational age at birth-pregnancyAssociation with PM_2.5_ during pregnancy and ASD with an OR of 1.15 (1.06, 1.24) per interquartile range (IQR) increase per 4.68 μg/m^3^Volk et al., 2013 [31]Case-control279 ASD cases and 245 controls from CaliforniaASD case classification based on the results from the ADOS (Autism Diagnostic Observation Schedule) and ADI-R (Autism Diagnostic Interview-Revised)Traffic-related air pollution was assigned to each mother’s location using a line-source air-quality dispersion modelLogistic additive models-child gender -child ethnicity -maximal education of parents -maternal age -prenatal smoking-pregnancy -trimester 1, 2, 3, -year 1Exposure to PM _2.5_, during pregnancy and during the first year of life was associated with ASD with an OR of 2.08 (95%CI, 1.93–2.25) Exposure to PM_10_ in the prenatal period was associated with ASD with an OR of 2.17 (1.49, 3.16) per each increase of 14.6 μg/m^3^ in IQR valueJo et al., 2019 [32]Cohort246,420 children from Southern CaliforniaASD diagnosis according to International Classification of Diseases, Ninth Revision, Clinical Modification (ICD-9-CM)PM_2.5_ measured at regulatory air monitoring stations was interpolated to estimate exposures during preconception and each pregnancy trimester, and first year of life at each child’s birth addressCox regression models-birth year -relevant maternal and child characteristics-preconception -trimester 1,2,3 -year 1Statistically significant association between exposures to PM_2.5_ and risk of ASD quantified by the following Hazard Ratios (HR) per each increase of 6.5 μg/m^3^ in IQR values in different phases of the pregnancy: 1.17 (95%CI 1.04–1.33) in the entire 9-months period; 1.10 (95%CI, 1.02–1.19) in the first trimester of pregnancy, 1.08 (95%CI, 1.00–1.18) in the third trimester of pregnancy; 1.21 (95%CI, 1.05–1.40) in the first year of lifePagalan et al., 2019 [33]CohortCohort of 129,439 children: 1276 were diagnosed with ASD from Metro Vancouver, British Columbia, CanadaASD case classification based on the results from the ADOS (Autism Diagnostic Observation Schedule) and ADI-R (Autism Diagnostic Interview-Revised)Monthly mean exposures to PM _2.5_, at the maternal residence during pregnancy were estimated with temporally adjusted, high-resolution land use regression modelsLogistic regression model-child sex -birth month -birth year -maternal age -maternal birthplace -neighborhood level urbanicity and income band-pregnancyNo significant association between PM_2.5_ and ASDGuxens et al., 2016 [34]Cohort8079 children from four European population-based birth/child cohorts (Sweden, Netherlands, Italy, and Spain)Autistic traits were assessed in children using:-the Autism Spectrum Disorder module of the Autism—Tics, Attention Deficit and Hyperactivity Disorders, and Other Comorbidities (A-TAC) inventory in the Swedish cohort-the Pervasive Developmental Problems (PDP) subscale of the Child Behavior Checklist for Toddlers in the Dutch cohort and in the Italian cohort-an adapted 18-item version of the Social Responsiveness Scale (SRS) in the Dutch cohort at age 6 years;-the Childhood Autism Spectrum Test (CAST) in the Spanish cohortsPM_2.5_ and PM_10_ absorbance were estimated for birth addresses by land-use regression models based on monitoring campaigns performed between 2008 and 2011. Levels of exposures were extrapolated back in time to exact pregnancy periodsLogistic regression model-age at delivery -educational level -country of birth -prenatal smoking -parity -maternal height -pre-pregnancy weight -pre-pregnancy body mass index -child’s sex -date of birth -child’s age at autistic trait assessment -information on the evaluator of the autistic traits -urbanity at child’s birth address-pregnancyNo significant association between PM_2.5_, PM_10_ and ASDKalkbrenner et al., 2015 [35]Case-Control979 ASD cases and 14,666 controls from North Carolina and San Francisco Bay Area in CaliforniaThe diagnosis of ASD was based on the Diagnostic and Statistical Manual of Mental Disorders, 4th Edition, Text Revision (DSM-IV-R)Exposure to PM_10_ at the birth address was assigned to each child by a geostatistical interpolation method using daily concentrations from air pollution regulatory monitorsLogistic generalized additive models-year -state -race -maternal education -maternal age -the calendar week of the child’s birth-pre-conception -trimester 1,2,3 -postnatalThe results obtained showed an association with the third trimester of pregnancy, with an OR of 1.36 (1.13, 1.63) per 10 μg/m^3^ increase in IQR valueYousefian et al., 2018 [36]Case-Control134 ASD cases and 388 controls from Tehran, IranThe diagnosis of ASD was based on the Diagnostic and Statistical Manual of Mental Disorders, 4th Edition, Text Revision (DSM-IV-R)Land-use regression models were used to estimate their annual mean exposure to ambient PM_10_Logistic regression-maternal age at birth -maternal education -paternal education -cousin marriage -maternal smoking during pregnancy -birth order -gestational age (weeks) -multiple births -maternal disease -paternal disease-pregnancyNo significant association between PM_10_ and ASDGong et al., 2017 [37]Case-Control5136 ASD cases 18,237 controls from SwedenThe diagnosis of ASD was based on the Diagnostic and Statistical Manual of Mental Disorders, 4th Edition, Text Revision (DSM-IV-R)Levels of PM _10_ from road traffic were estimated at residential addresses during mother’s pregnancy and the child’s first year of life by dispersion modelsLogistic regression model-municipality -calendar year of birth-pregnancy -first year of lifeNo significant association between PM_10_ and ASDJung et al., 2013 [39]Case-Control49,073 children from TaiwanASD diagnosis according to International Classification of Diseases, Ninth Revision, Clinical Modification (ICD-9-CM)Inverse distance weighting method was used to form exposure parameter for PM_10_Cox proportional hazards (PH) model-age -anxiety -gender -intellectual disabilities -preterm -SES-postnatalNo significant association between PM_10_ and ASD during the first for years of lifeGong et al., 2014 [38]Longitudinal cohort3426 twins born in Stockholm during 1992–2000The diagnosis of ASD was based on the Diagnostic and Statistical Manual of Mental Disorders, 4th Edition, Text Revision (DSM-IV-R)Residence time-weighted concentrations of PM_10_ from road traffic were estimated at participants’ addresses using dispersion modeling, controlling for seasonal variationMultivariate regression models-parity -gender -maternal age during pregnancy -maternal smoking during pregnancy -maternal marital status at birth year -parental education -family income -neighborhood deprivation at birth year-pregnancy -year 1 -year 9No significant association between PM_10_ and ASD

### 3.2. Association between NO_2_ and ASD

Several studies evaluated the association between maternal exposure to NO_2_ during pregnancy and in the first few years of the child’s life and the risk of the onset of ASD. In 2018, a study conducted on the Israeli population which enrolled 2098 cases and 54,191 controls, assessed maternal exposure during pregnancy and in the first nine months of the child’s life using dispersion models. The OR obtained through logistic regression was of 1.40 (95%CI: 1.09, 1.80) per IQR increase for 5.85 ppb showing that postnatal exposure to NO_2_ is associated with an increased risk of ASD diagnosis [40].

Again, the study of Ritz et al., already described above, evaluated also the possible relation between NO_2_ exposure and the increased risk of ASD. The data obtained showed that exposure to NO_2_ during the entire period of pregnancy increases the risk of a diagnosis of ASD. In particular, the authors estimated an adjusted OR of 1.08 (95%CI: 1.01, 1.15) for ASD per IQR increase for 11.41 μg/m^3^ in NO_2_ [26].

A 2018 study assessed the joint effects of taking folic acid in pregnancy and exposure to NO_2_ in association with the risk of diagnosing ASD. The survey recruited the CHARGE study participants (346 cases and 260 controls). This study showed that taking folic acid during pregnancy reduces the risk of ASD among those exposed to high NO_2_ levels in the prenatal period [41].

In the aforementioned study by Volk et al., which examined the relationship between traffic-related air pollution and ASD, the exposure to NO_2_ was also investigated. The results confirmed an association in the prenatal period and the risk of ASD with an OR of 1.81 (95%CI, 1.37, 3.09) [31].

In the same year, a Californian case-control study, previously described, assessed the influence of exposure to environmental pollutants deriving from vehicular traffic during pregnancy on the onset of the ASD. The results obtained highlighted an association between the development of ASD and prenatal exposure to NO_2_ [30].

However, a Swedish case-control study did not find clear associations between NO_2_ exposures during pregnancy, the first year of life, and the first nine years of life and ASD [38].

Several cohort studies were also conducted to evaluate the association between maternal exposure to NO_2_ during pregnancy and in the very early years of the baby’s life and the risk of ASD onset.

In 2019 a cohort study was conducted on a Swedish population consisting of 48,571 children born between 1999 and 2009. Exposure to nitrogen oxides (NOx), of which NO_2_ is the largest component, was assessed using Gaussians dispersion models during the gestational period considering maternal residence. Researchers found associations between NOx exposure during the prenatal period and an increased risk of diagnosis for ASD [42].

Another cohort study conducted in 2013 in Taiwan involved 49,073 children aged up to three years old. Inverse distance weighting method was used to describe exposure parameter for several pollutants, including NO_2_. A time-dependent Cox proportional hazards model was used to assess the associations between the average exposure to air pollutants of previous years and newly diagnosed ASD children. The study provided evidence of association between NO_2_ in the first four years of life and an increased risk of ASD [39].

In contrast to the evidence gathered from these studies, two further cohort studies found no statistically significant associations. Actually, a 2019 Canadian cohort study involving 132,256 children born between 2004 and 2009 (with 1307 who had received an ASD diagnosis) found no significant association between NO_2_ and ASD [33]. Moreover, the study carried out by Guxens et al. (already described above) involving four European countries, assessed also the maternal exposure to NO_2_ during different periods of pregnancy and its putative relationship with ASD onset without finding out significant evidence [34]. Results for associations between NO_2_ and ASD are summarized in Table 2.

### 3.3. Association between O_3_ and ASD

The association between O_3_ exposure and ASD onset was investigated in the aforementioned study of Mc Guinn et al. with a positive correlation being identified. There was a variation by exposure time period for the O_3_–ASD relation, with a stronger association observed during the third trimester of pregnancy and an OR of 1.2 (95%CI: 1.1, 1.4) per 6.6 ppb increase in O_3_ [22].

The Californian study by Becerra et al.—which examined the influence of air pollution on ASD onset by using LUR model for exposure estimations—highlighted an association between ASD and prenatal exposure to O_3_. A 12–15% relative increase in odds of autism for O_3_ was estimated, with an OR of 1.12 (95%CI: 1.06, 1.19) per 11.54 ppb increase in O_3_ [30].

Another study investigated the link between ASD and the O_3_ concentrations to which mothers were exposed during pregnancy and for the first two years of the child’s life. The results showed an association with the exposure to high levels of O_3_ during the second year of life [25].

A cohort study, performed in Taiwan in 2013, generated positive evidence about the association between O_3_ exposure during the first four years of life and an increased risk of ASD diagnosis [39]. However, another cohort study carried out in California in 2019 did not found significant association [32].

Studies investigating the association between O_3_ and ASD are summarized in Table 3.

### 3.4. Association between SO_2_ and ASD

In literature, the association between maternal exposure to SO_2_ during pregnancy and the earliest years of children’s life and the risk of ASD onset has been poorly investigated. The study performed by Ritz et al. examined the exposure—at mothers’ houses—to different environmental pollutants, among which SO_2_, during the gestation and in the following nine months by using the AirGIS dispersion model. The authors estimated an OR of 1.21 (95%CI: 1.13, 1.29) for ASD onset per each increase of 2.8 μg/m^3^ in SO_2_ for IQR value, nine months after pregnancy [26].

Similarly, a Taiwanese cohort study showed an association between the exposure to SO_2_ in the first four years of life and the risk of ASD diagnosis. The researchers highlighted a 17% of increased risk for ASD per each increase of 1 ppb in SO_2_ levels (95%CI: 1.09, 1.27) [39].

At the opposite, the study carried out by Yousefian et al. performed in Teheran in 2018 did not found statistically significant associations between the mean level of exposure to SO_2_ and the risk of developing ASD during childhood [36].

Results of associations between SO_2_ and ASD are reported in Table 4.

## 4. Discussion

This review examined a total of 20 studies investigating the association between maternal and early life exposure to air pollutants and ASD in children, addressing the specific burden attributable to different air pollutants (PM_2.5_, PM_10_, NO_2_, O_3_, and SO_2_).

We found out a marked association between maternal exposure to PM during pregnancy or in the first years of the children’s life and the risk of subsequent ASD onset. In turn, this association was found to be stronger with PM_2.5_ (confirmed in nine case-control studies and three cohort studies) while the association between PM_10_ and ASD was less evident. However, our analysis showed that half of eight case-control studies selected found a positive association between maternal exposure to PM_10_ during the pregnancy (particularly in the third trimester) or in the early years of life and ASD onset [26,27,31,35]. The remaining four case-control studies did not find the same association [28,36,37,39] as well as the two cohort studies selected in the present review [34,38]. Another air pollutant that has been extensively studied and related to ASD is NO_2_. The six case-control studies that analyzed the association between maternal or early life exposure to NO_2_ and ASD onset showed a positive association [26,30,31,39,40,41]. Moreover, only one out of four cohort studies [33,34,38,42] analyzing this kind of relationship reported a positive association between NO_2_ exposure during pregnancy or early life and ASD [42].

All the four case-control studies considered in this review generated evidence about the association between maternal or early years of life exposure to O_3_ and ASD [23,25,30,39].

The evidence, however, is limited with regard to SO_2_, due to the small number of studies conducted. Two out of three studies reported a positive association between maternal or early life exposure to SO_2_ and the onset of ASD [26,39].

In summary, while there is some evidence indicating that maternal exposure to PM_2.5_ may increase the risk of ASD, the effect of PM_10_, O_3_, NO_2_, and SO_2_ on ASD seem to be weaker or limited. An increase in the number of studies for a future meta-analysis would improve the statistical power to identify associations between air pollutants and ASD.

It would also be interesting to understand the possible contribution not only of individual pollutants but also the cumulative burden of mixture of pollutants, usually contaminating environment, food chain, and possibly the amniotic liquid or placenta. Therefore, future works should include improved spatiotemporal estimates of exposure to air toxics, considering computational fluid-dynamics models on community dwelling population during daily life.

A very interesting aspect that emerged from the analysis of the studies considered in this review is the critical “time window”, which is the period when exposure to environmental pollutants seems to be most incisive. This period corresponds to pregnancy; indeed, autistic symptoms commonly appear in the second and sometimes the first year of life, indicating a strong likelihood of origins in the prenatal and early postnatal periods. Neurogenesis takes place at an astonishing rate, averaging 250,000 new neurons per minute during gestation resulting in 100 billion neurons at birth [43]. Most of the neuronal growth occurs in the third trimester, when 40,000 synapses are formed per minute. Moreover, brain development comprises an array of qualitatively different processes with overlapping timing: neural tube formation, cell proliferation and differentiation, migration, dendritic arborization, synaptogenesis, apoptosis, formation, and connectivity of cortical mini-columns, and myelination. The synaptogenesis is known to have a major role in ASD, as it accelerates in the middle of the second trimester (but continues until adolescence), whereas neural migration is completed by birth [44].

Evidence on the association between air pollution and neurodevelopmental disorders dramatically increased in the last years. Many studies investigated the risk of developing, in addition to ASD, other disorders such as learning disabilities, attention deficit and hyperactivity disorder (ADHD) when children (or their mother) are exposed to high level of airborne pollutants, showing a possible link [45]. The long-term exposure to air pollution begin in the prenatal period and continue during the lifetime, causing the possible onset of chronic adverse outcome such as cancer, cardiovascular and respiratory diseases [46]. Recently, also the cognitive decline in the elderly was associated with atmospheric particles and chemicals [47]. In particular, older adults living in polluted areas showed a decrease in cognitive function and may be at greater risk of experiencing progressive neurodegenerative pathologies such as Alzheimer’s disease [48].

Not only the brain, but also another fundamental organ for the regulation of fetal development, such as the placenta, may be influenced by environmental factors. Placenta can be considered as an intermediate matrix having the potential to express, in association with prenatal exposure to air pollution, distinct biological signatures which may be useful as early-life markers of disease development later in life [49]. Indeed, several studies showed that during pregnancy the exposure to an increased level of PM (PM_2.5_ or PM_10_) was associated with changes in placental epigenetic markers (mainly DNA methylation), transcriptomic, and proteomic biomarkers which should be extensively studied to understand the possible consequences on fetus health and adult life [49].

Although the results obtained from the studies analyzed in this review seem to confirm that some airborne environmental pollutants can play an important role in the complex etiology of ASD, they represent only a part of the search for environmental causative agents determining ASD. This kind of investigation must take into account a wider range of exposures concerning contaminants in the air, water, soil, and food. The studies analyzed in our review are moving in this direction, since other variables that influence the biology and, therefore, the health was taken into consideration: nutrition, smoking, financial, educational, and social aspects, the structure of the family, living place (urban/suburban/rural environment), workplaces, and microbiome. Unlike the inherited nuclear DNA, these exposures represent modifiable factors, thus opening the door to preventive interventions at various levels. In an epigenetic perspective of the research on autism, it is crucial to reduce the exposures to environmental risk factors [14].

Another field of investigation may be the comparison of the incidence and prevalence rates of ASD between a highly polluted area, where it is assumed that there may be a greater risk of occurrence of ASD, and a control area, in order to understand how cases are distributed in the two areas. Moreover, it would also be interesting to investigate if children born during the spring of 2020, when lockdown measures, due to the spread of SARS-CoV-2, resulted in decreased PM_2.5_ levels in highly polluted cities, will lead to a decrease in ASD cases compared to the years 2018–2019.

What emerged from the comparison of the studies we selected in this review was the great heterogeneity of the results, which could in part be due to the different number of children recruited, the various statistical methods used for data analysis, and the different ASD assessment systems among studies. The evidence from the case-control studies appeared to be more solid than that from the cohort studies and this could be explained by analyzing the different methods of assessing exposure to air pollution and confounding factors that represented potential bias in some case-control studies.

Confounding due to lifestyle factors, such as smoking in pregnancy, including active smoking and passive smoking, could be problematic, as this factor might be directly associated with an increased risk of the onset of ASD, as reported in several studies [50,51].

Another potential confounding factor could be that related to socio-economic position at the individual and residential area level. In this context, Miranda et al. found that poorer and minority neighborhoods were more likely to experience higher levels of pollution and were also highly correlated with an increased burden of disease. Despite the important need to adjust to these confounding factors, not all of the studies we reviewed corrected or examined the modification of effects based on the socioeconomic position of individuals or their communities [52].

Since it is necessary to further establish the direct associations between air pollution and adverse neurological effects, such as the onset of ASD, the assessment of exposure to air pollution is a crucial element for future studies. Many studies used a binary exposure measurement, i.e., they compared a high-pollution area versus a low-pollution area by measuring the annual concentrations of air pollutants from the surrounding monitoring stations or taking into account distance from traffic routes as an estimate of atmospheric pollution. These exposure measurements can lead to incorrect exposure classification and, for this reason, it is crucial to develop more accurate methods for measuring chronic exposure to air pollution in this field.

To get a more precise estimate of air pollution exposure, the most accurate method would be to evaluate and quantify individual exposures. Personal monitoring could provide an exposure estimate that is less prone to misclassification than other methods of measuring exposure to air pollution, especially when taking into account well-defined time windows such as the first 1000 days of the life of the child, including nine months of gestation and the first two years of the child’s life. However, this is more difficult to do if we examine a large cohort of subjects, so this motivation could explain the lower evidence found in the cohort studies.

In this context, a LUR model, can integrate personal monitoring and biomonitoring methods including GIS parameters, such as traffic density, population density, which are used to predict small-scale spatial variation of pollutants [53,54]. However, this approach has limitations regarding the estimation of air pollution levels referred to the address of residence as ignores historical exposures since people are constantly on the move and are not confined to the place where the exposure was assessed bringing to a potential misclassification of exposure.

In order to reduce the potential for misclassification of exposure, efforts should be made to include information on individual residency history and length of stay in LUR templates.

In general, the studies we included in our review had critical points, such as the recruitment of small cohorts, the poor comparisons among different areas characterized by different kinds of pollution, the lack of use of a standardized statistical method, the use of different models for assessing the level of exposure to pollutants, and the difficulty of quantifying the exposure both for single molecules or their mixture. Furthermore, air pollution is a complex mixture of toxins for which the biological effects on the development of ASD of individual agents are hard to identify considering also the synergistic effects that the various pollutants could have. Therefore, further studies must take into consideration samples to be compared to many areas, ensuring that the participants were exposed to a wide range of pollutant concentrations with individual-level exposure measurements to multiple compounds by various pathways (air, water, and diet), combined with genetic information.

In addition, the use of exposure biomarkers—which are able to highlight a biological alteration caused by air pollution—could represent a more effective method for measuring these exposures and can be combined with the direct detection of pollutants. Indeed, the subjects’ response to air pollution can be variable and depends on individual susceptibility.

In this way, it would be possible to identify people who actually suffered damages caused by airborne chemicals and this data could be correlated with the risk of ASD. For example, micronuclei could be considered, as they are biomarkers of early biological effect which can detect the presence of initial (and reversible) alterations in the chromosomal structure and oxidative damage to DNA caused by air pollution exposure [55,56].

Moreover, in order to quantify individual exposures to specific environmental pollutants, further studies should examine also epigenetic alterations which determine changes in gene expression without altering DNA sequence [57]. These modifications, which include DNA methylation, histone modifications, and microRNAs (miRNAs) expression, operate together in a synergic manner, influencing genome expression patterns and functions in response to exogenous stimuli or exposure [58,59,60].

Biomarkers that reflect specific exposures have the potential to measure the real integrated internal dose from all routes of complex environmental exposures. In particular, miRNAs have been studied as biomarkers in various diseases and have shown potential as environmental exposure biomarkers [61,62]. Several studies have suggested that environmental factors may interact with the genetic factors to increase the risk of ASD [10,63,64,65]. Although today there are still no clear associations between specific miRNAs and specific pollutants, these epigenetic markers are promising and could represent one of those factors explaining the link between genetics and environment [66].

## 5. Conclusions

Air pollution should be considered among the emerging risk factors for ASD. The relationship identified between airborne pollutants and ASD in the studies analyzed in this review must necessarily be confirmed and implemented by further epidemiological studies with more personalized assessment of indoor and outdoor exposure measurements (i.e., through biomarkers of exposure), including more confounders, and using both single pollutant and multiple pollutant statistical models. Moreover, also toxicological studies are needed to understand which molecular pathways are involved in the development of ASD and whether they may result altered after the exposure to specific pollutants. All these efforts will be necessary to determine whether there are causal associations with air pollution and ASD onset or not.

## Figures and Tables

**Figure 1 ijerph-18-01204-f001:**
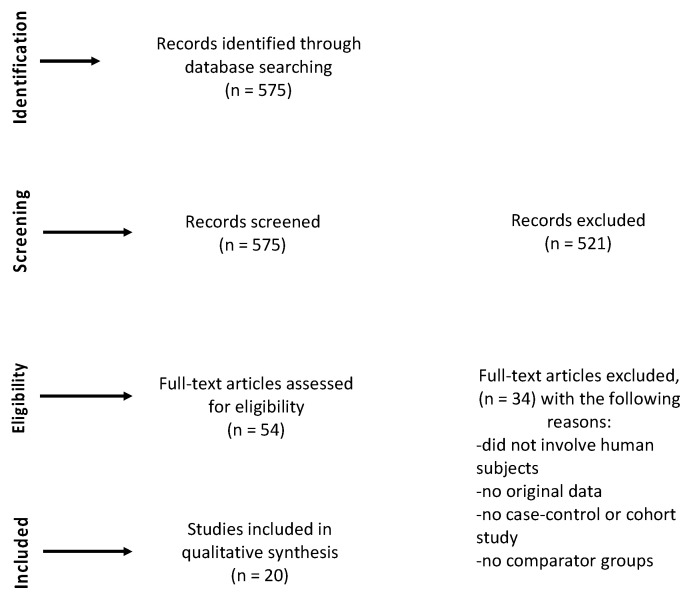
Flow diagram.

**Table 2 ijerph-18-01204-t002:** Studies investigating the association between exposure to NO_2_ and ASD development.

Authors and Year	Study Design	Sample Size/Country	ASD Outcome Assessment	Exposure Assessment	Method of Analysis	Adjustment Variables	Time Window of Exposure	Main Findings
Ritz et al., 2018 [26]	Case-control	15,387 ASD cases and 68,139 controls from Denmark	ASD diagnosis according to International Classification of Diseases, Tenth Revision, Clinical Modification (ICD-10-CM)	Exposures estimates provided by the dispersion models (AirGIS) for the detected pollutants (NO_2_, SO_2_, PM_2.5_ and PM_10_) in relation to the maternal residence during the examined periods	Conditional logistic regression	-parental age -neighborhood -socio-economic indicators -maternal smoking	-pregnancy -9 months after pregnancy	The data obtained showed that exposure during the entire period of pregnancy to NO_2_ increases the risk of a diagnosis of ASD. In particular, the authors estimated an adjusted ORs for ASD per IQR increase for 11.41 μg/m^3^ with NO_2_ of 1.08 (95%CI: 1.01, 1.15)
Becerra et al., 2013 [30]	Case-control	7603 ASD cases and 10 controls per case from Los Angeles, California	The diagnosis of ASD was based on the Diagnostic and Statistical Manual of Mental Disorders, 4th Edition, Text Revision (DSM-IV-R)	Modeled concentrations of PM_2.5_ from air monitoring station 1993–2006, assigned by residential address at delivery/birth	Conditional logistic regression	-maternal age -maternal place of birth -race -education -type of birth -parity -insurance type -gestational age at birth	-pregnancy	The results obtained highlighted an association between the development of ASD and prenatal exposure to NO_2_
Volk et al., 2013 [31]	Case-control	279 ASD cases and 245 controls from California	ASD case classification based on the results from the ADOS (Autism Diagnostic Observation Schedule) and ADI-R (Autism Diagnostic Interview-Revised)	Traffic-related air pollution was assigned to each mother’s location using a line-source air-quality dispersion model	Logistic additive models	-child gender -child ethnicity -maximal education of parents -maternal age -prenatal smoking	-pregnancy -trimester 1, 2, 3, -year 1	The results confirmed an association in the prenatal period and the risk of ASD with an OR of 1.81
Pagalan et al., 2019 [33]	Cohort	Cohort of 129,439 children: 1276 were diagnosed with ASD from Metro Vancouver, British Columbia, Canada	ASD case classification based on the results from the ADOS (Autism Diagnostic Observation Schedule) and ADI-R (Autism Diagnostic Interview-Revised)	Monthly mean exposures to PM_2.5_, at the maternal residence during pregnancy were estimated with temporally adjusted, high-resolution land use regression models	Logistic regression model	-child sex -birth month -birth year -maternal age -maternal birthplace -neighborhood level urbanicity and income band.	-pregnancy	No significant association between NO_2_ and ASD
Guxens et al., 2016 [34]	Cohort	8079 children from four European population-based birth/child cohorts (Sweden, Netherlands, Italy, and Spain)	Autistic traits were assessed in children using: -the Autism Spectrum Disorder module of the Autism—Tics, Attention Deficit and Hyperactivity Disorders, and Other Comorbidities (A-TAC) inventory in the Swedish cohort -the Pervasive Developmental Problems (PDP) subscale of the Child Behavior Checklist for Toddlers in the Dutch cohort and in the Italian cohort -an adapted 18-item version of the Social Responsiveness Scale (SRS) in the Dutch cohort at age 6 years; -the Childhood Autism Spectrum Test (CAST) in the Spanish cohorts	PM_2.5_ absorbance was estimated for birth addresses by land-use regression models based on monitoring campaigns performed between 2008 and 2011. Levels of exposures were extrapolated back in time to exact pregnancy periods	Logistic regression model	-age at delivery -educational level -country of birth -prenatal smoking -parity -maternal height -pre-pregnancy weight -pre-pregnancy body mass index -child’s sex -date of birth -child’s age at autistic trait assessment -information on the evaluator of the autistic traits -urbanity at child’s birth address	-pregnancy	No significant association between NO_2_ and ASD
Jung et al., 2013 [39]	Case-Control	49,073 children from Taiwan	ASD diagnosis according to International Classification of Diseases, Ninth Revision, Clinical Modification (ICD-9-CM)	Inverse distance weighting method was used to form exposure parameter for PM_10_	Cox proportional hazards (PH) model	-age -anxiety -gender -intellectual disabilities -preterm -SES	-postnatal	The study provided evidence of association between NO_2_ in the first 4 years of life and an increased risk of ASD
Gong et al., 2014 [38]	Longitudinal cohort	3426 twins born in Stockholm during 1992–2000	The diagnosis of ASD was based on the Diagnostic and Statistical Manual of Mental Disorders, 4th Edition, Text Revision (DSM-IV-R)	Residence time-weighted concentrations of PM_10_ from road traffic were estimated at participants’ addresses using dispersion modeling, controlling for seasonal variation	Multivariate regression models	-parity -gender -maternal age during pregnancy -maternal smoking during pregnancy -maternal marital status at birth year -parental education -family income -neighborhood deprivation at birth year	-pregnancy -year 1 -year 9	No significant association between NO_2_ and ASD
Goodrich et al., 2018 [41]	Case-Control	346 ASD cases and 260 controls from California	ASD case classification based on the results from the ADOS (Autism Diagnostic Observation Schedule) and ADI-R (Autism Diagnostic Interview-Revised)	Estimates of exposure to near roadway air pollution (NRP) and criteria air pollutant measures were assigned based on maternal residential history	Logistic regression model	-self-reported FA intake for each month of pregnancy	-pregnancy -trimester 1, 2, 3	This study showed that taking folic acid during pregnancy reduces the risk of ASD among those exposed to prenatal high NO_2_ levels
Raz et al., 2018 [40]	Case-Control	2098 ASD cases and 54,191 controls from Israel	The diagnosis of ASD was based on the Diagnostic and Statistical Manual of Mental Disorders, 4th Edition, Text Revision (DSM-IV-R)	NO_2_ Exposure was based on an optimized dispersion model	Logistic regression	-year of birth -calendar month of birth -population group -paternal age -socioeconomic status	-pre-conception -pregnancy -9 months after birth	Postnatal exposure to NO_2_ is associated with an increased risk of ASD diagnosis with values of 1.40 (95%CI: 1.09, 1.80) per IQR increase for 5.85 ppb
Oudin et al., 2019 [42]	Longitudinal cohort	48,571 children born between 1999 and 2009 in southern Sweden	ASD diagnosis according to International Classification of Diseases, Tenth Revision, Clinical Modification (ICD-10-CM)	Modelled nitrogen oxide (NO_x_) levels derived from a Gaussian dispersion model	Logistic regression model	-maternal residency during pregnancy -perinatal factors collected from a regional birth registry -socio-economic factors	-pregnancy	Positive associations between NO_x_ exposure during pregnancy and ASD with an adjusted Odds Ratio (OR) of 1.40 (95%CI: 1.02–1.93)

**Table 3 ijerph-18-01204-t003:** Studies investigating the association between exposure to O_3_ and ASD development.

Authors and Year	Study Design	Sample Size/Country	ASD Outcome Assessment	Exposure Assessment	Method of Analysis	Adjustment Variables	Time Window of Exposure	Main Findings
McGuinn et al., 2020 [23]	Case-control	674 ASD cases and 855 controls fromCalifornia, Colorado, Georgia, Maryland, North Carolina, and Pennsylvania	ASD case classification based on the results from the ADOS (Autism Diagnostic Observation Schedule) and ADI-R (Autism Diagnostic Interview-Revised)	Satellite-based model to assign air pollutant exposure (PM_2.5_, O_3_) averages during several critical periods of neurodevelopment	Logistic regression model	-study site -maternal age -maternal education -maternal race -maternal smoking, -month and year of birth	-3 months before pregnancy -each trimester of pregnancy -the entire pregnancy -first year of life	There was a variation by exposure time period for the O_3_–ASD relation, with a stronger association observed during the third trimester of pregnancy and an OR of 1.2 (95%CI: 1.1, 1.4) per 6.6 ppb increase in O_3_
Kaufman et al., 2019 [25]	Case-control	428 ASD cases and 6420 controls from metropolitan Cincinnati area in southwest Ohio	ASD diagnosis according to International Classification of Diseases, Ninth Revision, Clinical Modification (ICD-9-CM)	Daily PM_2.5_ individual exposure estimations provided by the United States Environmental Protection Agency (US EPA), based on their residential address	Logistic regression model	-maternal and birth-related confounders -multiple temporal exposure windows	-each trimester pregnancy -first year of life -second year of life	Positive association with O_3_ exposure during the 2nd year of life (OR range across categories: (1.29–1.42)
Becerra et al., 2013 [30]	Case-control	7603 ASD cases and 10 controls per case from Los Angeles, California	The diagnosis of ASD was based on the Diagnostic and Statistical Manual of Mental Disorders, 4th Edition, Text Revision (DSM-IV-R)	Modeled concentrations of PM_2.5_ from air monitoring station 1993–2006, assigned by residential address at delivery/birth	Conditional logistic regression	-maternal age -maternal place of birth -race -education -type of birth -parity -insurance type -gestational age at birth	-pregnancy	Positive association with O_3_ during pregnancy and ASD. 1.12 (95%CI, 1.06, 1.19) per 11.54 ppb increase in O_3_
Jo et al., 2019 [32]	Cohort	246,420 children from Southern California	ASD diagnosis according to International Classification of Diseases, Ninth Revision, Clinical Modification (ICD-9-CM)	PM_2.5_ measured at regulatory air monitoring stations was interpolated to estimate exposures during preconception and each pregnancy trimester, and first year of life at each child’s birth address	Cox regression models	-birth year -relevant maternal and child characteristics	-preconception -trimester 1,2,3 -year 1	No significant association between O_3_ and ASD
Jung et al., 2013 [39]	Case-Control	49,073 children from Taiwan	ASD diagnosis according to International Classification of Diseases, Ninth Revision, Clinical Modification (ICD-9-CM)	Inverse distance weighting method was used to form exposure parameter for PM_10_	Cox proportional hazards (PH) model	-age -anxiety -gender -intellectual disabilities -preterm -SES	-postnatal	The study suggested an association between O_3_ exposure in the first for years of life and ASD

**Table 4 ijerph-18-01204-t004:** Studies investigating the association between exposure to SO_2_ and ASD development.

Authors and Year	Study Design	Sample Size/Country	ASD Outcome Assessment	Exposure Assessment	Method of Analysis	Adjustment Variables	Time Window of Exposure	Main Findings
Ritz et al., 2018 [26]	Case-control	15,387 ASD cases and 68,139 controls from Denmark	ASD diagnosis according to International Classification of Diseases, Tenth Revision, Clinical Modification (ICD-10-CM)	Exposures estimates provided by the dispersion models (AirGIS) for the detected pollutants (NO_2_, SO_2_, PM_2.5_ and PM_10_) in relation to the maternal residence during the examined periods	Conditional logistic regression	-parental age -neighborhood -socio-economic indicators -maternal smoking	-pregnancy -9 months after pregnancy	The authors estimated an OR of 1.21 (95%CI: 1.13, 1.29) for ASD onset per each increase of 2.8 μg/m^3^ SO_2_ in IQR value, 9 months after pregnancy
Yousefian et al., 2018 [36]	Case-Control	134 ASD cases and 388 controls from Tehran, Iran	The diagnosis of ASD was based on the Diagnostic and Statistical Manual of Mental Disorders, 4th Edition, Text Revision (DSM-IV-R)	Land-use regression models were used to estimate their annual mean exposure to ambient PM_10_	Logistic regression	-maternal age at birth -maternal education -paternal education -cousin marriage -maternal smoking during pregnancy -birth order -gestational age (weeks) -multiple births -maternal disease -paternal disease	-pregnancy	No significant association between SO_2_ and ASD
Jung et al., 2013 [39]	Case-Control	49,073 children from Taiwan	ASD diagnosis according to International Classification of Diseases, Ninth Revision, Clinical Modification (ICD-9-CM).	Inverse distance weighting method was used to form exposure parameter for PM_10_	Cox proportional hazards (PH) model	-age -anxiety -gender -intellectual disabilities -preterm -SES	-postnatal	The researchers highlighted a 17% increased risk for ASD per each increase of 1 ppb in SO_2_ levels (95%CI 1.09–1.27)

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
