# Peer review of "Early-Life Exposure to Environmental Air Pollution and Autism Spectrum Disorder: A Review of Available Evidence"

_ijerph, 2021, doi:10.3390/ijerph18031204_

Round 1

Reviewer 1 Report

This review offers a brief summary of relevant studies investigated the relation between exposure to air pollutants (P2.5, PM10, NO2, SO2) in utero and/or in the first year(s) of life and the development of symptoms of Autism Spectrum Disorder (ASD).

The tables are nice and informative but somewhat redundant as the same study by Ritz et al. appears in all 4 tables (one table per pollutant). I especially don't see the point of separating PM2.5 and PM10, as PM10 contains PM2.5 and both contain nanoparticles (PM<1um), which are the only particles able to cross the brain barrier. Thus, contrasting results between PM2.5 and PM10 should be reported side by side. 

My main issue with the result section and the discussion is the lack of critical assessment of the strength and weakness of these studies. At first glance: some studies have a 100 time more ASD cases than others; not all studies control for maternal smoking; not all studies define autism the same way; most studies use different statistical approaches... What is the authors opinion about which studies are better. It is also striking that in contrast to case control studies, birth cohorts struggle to find any significant association between air pollutants and ASD. The authors should discuss potential explanations and look into potential bias in some of the case control studies.

The authors should mention that associations do not prove causation and suggest ways by which future studies could strengthen the quality of their studies. For examples more personalized assessment of indoor and outdoor exposure measurements, inclusion of more confounders, using both single pollutant and multiple pollutant statistical models... 

It would also be interesting to investigate if children born during the spring of 2020, when lockdown measures resulted in decreased PM2.5 levels in highly polluted cities will lead to a decrease in ASD cases compared to 2018-19.

The mention of MicroRNAs as biomarkers of environmental exposure is interesting and warrants more research but there is a long road ahead before a specific MicroRNA can be linked to a specific pollutant. 

The authors may want to mention in a paragraph in the discussion published findings about the presence of traffic related particulate matter in the brain as well as in the placenta and fetus (of animal and/or humans). This might help: Luyten LJ, et al. Air pollution and the fetal origin of disease: a systematic review of the molecular signatures of air pollution exposure in human placenta. Environ Res. 2018;166:310–323.

Finally, the authors should put these studies focussing on ASD into the larger context of air pollution and learning disabilities as well as cognitive decline in the elderly.

Author Response

Dear Reviewer 1,

thank you so much for your valuable comments and criticisms on our manuscript, which are helpful for revising and improving our paper. The paper titled as "Early-life exposure to environmental air pollution and Autism Spectrum Disorder: a review of available evidence " (ijerph-1023518), by Giovanni Imbriani, Alessandra Panico, Tiziana Grassi, Adele Idolo, Francesca Serio, Francesco Bagordo, Giovanni De Filippis, Donato De Giorgi, Gianfranco Antonucci, Prisco Piscitelli, Manuela Colangelo, Luigi Peccarisi, Maria Rosaria Tumolo, Roberto De Masi, Alessandro Miani and Antonella De Donno, has been reviewed according to reviewers recommendations using the "Track Changes" function in Microsoft Word.

 The comments and the corrections are also outlined below.

Reviewer 1

This review offers a brief summary of relevant studies investigated the relation between exposure to air pollutants (P2.5, PM10, NO2, SO2) in utero and/or in the first year(s) of life and the development of symptoms of Autism Spectrum Disorder (ASD).

  1. The tables are nice and informative but somewhat redundant as the same study by Ritz et al. appears in all 4 tables (one table per pollutant). I especially don't see the point of separating PM2.5 and PM10, as PM10 contains PM2.5 and both contain nanoparticles (PM<1um), which are the only particles able to cross the brain barrier. Thus, contrasting results between PM2.5 and PM10 should be reported side by side.
  2. We have merged and integrated data on PM2.5and PM10in the results section and in the table 1. In this way we have also reduced the redundancy of information in tables as suggested by the reviewer.

  1. My main issue with the result section and the discussion is the lack of critical assessment of the strength and weakness of these studies. At first glance: some studies have a 100 time more ASD cases than others; not all studies control for maternal smoking; not all studies define autism the same way; most studies use different statistical approaches... What is the authors opinion about which studies are better. It is also striking that in contrast to case control studies, birth cohorts struggle to find any significant association between air pollutants and ASD. The authors should discuss potential explanations and look into potential bias in some of the case control studies.
  2. We have discussed the heterogeneity in findings and all the valuable comments of the reviewer at lines 727-768: “What emerged from the comparison of the studies we selected in this review was the great heterogeneity of the results, which could in part be due to the different number of children recruited, the various statistical methods used for data analysis, and the different ASD assessment systems among studies. The evidence from the case-control studies appeared to be more solid than that from the cohort studies and this could be explained by analyzing the different methods of assessing exposure to air pollution and confounding factors that represented potential bias in some case-control studies.

Confounding due to lifestyle factors, such as smoking in pregnancy, including active smoking and passive smoking, could be problematic, as this factor might be directly associated with an increased risk of the onset of ASD, as reported in several studies [50, 51]. Another potential confounding factor could be that related to socio-economic position at the individual and residential area level. In this context, Miranda et al. found that poorer and minority neighborhoods were more likely to experience higher levels of pollution and were also highly correlated with an increased burden of disease. Despite the important need to adjust to these confounding factors, not all of the studies we reviewed corrected or examined the modification of effects based on the socioeconomic position of individuals or their communities [52].

Since it is necessary to further establish the direct associations between air pollution and adverse neurological effects, such as the onset of ASD, the assessment of exposure to air pollution is a crucial element for future studies. Many studies used a binary exposure measurement, i.e. they compared a high-pollution area versus a low-pollution area by measuring the annual concentrations of air pollutants from the surrounding monitoring stations or taking into account distance from traffic routes as an estimate of atmospheric pollution. These exposure measurements can lead to incorrect exposure classification and, for this reason, it is crucial to develop more accurate methods for measuring chronic exposure to air pollution in this field.

To get a more precise estimate of air pollution exposure, the most accurate method would be to evaluate and quantify individual exposures. Personal monitoring could provide an exposure estimate that is less prone to misclassification than other methods of measuring exposure to air pollution, especially when taking into account well-defined time windows such as the first 1000 days of the life of the child, including nine months of gestation and the first 2 years of the child's life. However, this is more difficult to do if we examine a large cohort of subjects, so this motivation could explain the lower evidence found in the cohort studies.

In this context, a LUR model, can integrate personal monitoring and biomonitoring methods including GIS parameters, such as traffic density, population density, which are used to predict small-scale spatial variation of pollutants [53, 54].  However, this approach has limitations regarding the estimation of air pollution levels referred to the address of residence as ignores historical exposures since people are constantly on the move and are not confined to the place where the exposure was assessed bringing to a potential misclassification of exposure.

In order to reduce the potential for misclassification of exposure, efforts should be made to include information on individual residency history and length of stay in LUR templates.”

  1. The authors should mention that associations do not prove causation and suggest ways by which future studies could strengthen the quality of their studies. For examples more personalized assessment of indoor and outdoor exposure measurements, inclusion of more confounders, using both single pollutant and multiple pollutant statistical models...
  2. We have considered these aspects in the Discussion section and we have also reported a brief consideration in the Conclusions: “The relationship identified between airborne pollutants and ASD in the studies analyzed in this review must necessarily be confirmed and implemented by further epidemiological studies with more personalized assessment of indoor and outdoor exposure measurements (i.e. through biomarkers of exposure), including more confounders, and using both single pollutant and multiple pollutant statistical models. Moreover, also toxicological studies are needed to understand which molecular pathways are involved in the development of ASD and whether they may result altered after the exposure to specific pollutants. All these efforts will be necessary to determine whether there are causal associations with air pollution and ASD onset or not.

  1. It would also be interesting to investigate if children born during the spring of 2020, when lockdown measures resulted in decreased PM2.5 levels in highly polluted cities will lead to a decrease in ASD cases compared to 2018-19.
  2. We have included this interesting suggestion in the Discussion section at lines 725-728.

  1. The mention of MicroRNAs as biomarkers of environmental exposure is interesting and warrants more research but there is a long road ahead before a specific MicroRNA can be linked to a specific pollutant.
  2. We specified this consideration at lines 793-795: “Although today there are still no clear associations between specific miRNAs and specific pollutants, these epigenetic markers are promising and could represent one of those factors explaining the link between genetics and environment [55].

  1. The authors may want to mention in a paragraph in the discussion published findings about the presence of traffic related particulate matter in the brain as well as in the placenta and fetus (of animal and/or humans). This might help: Luyten LJ, et al. Air pollution and the fetal origin of disease: a systematic review of the molecular signatures of air pollution exposure in human placenta. Environ Res. 2018; 166:310–323.
  2. Considering the systematic review indicated by the review we have discussed these issues at lines 683-710: “Evidence on the association between air pollution and neurodevelopmental disorders dramatically increased in the last years. Many studies investigated the risk of developing, in addition to ASD, other disorders such as learning disabilities, attention deficit and hyperactivity disorder (ADHD) when children (or their mother) are exposed to high level of airborne pollutants, showing a possible link [45]. The long-term exposure to air pollution begin in the prenatal period and continue during the lifetime, causing the possible onset of chronic adverse outcome such as cancer, cardiovascular and respiratory diseases [46]. Recently, also the cognitive decline in the elderly was associated with atmospheric particles and chemicals [47]. In particular, older adults living in polluted areas showed a decrease in cognitive function and may be at greater risk of experiencing progressive neurodegenerative pathologies such as Alzheimer’s disease [48].

Not only the brain, but also another fundamental organ for the regulation of fetal development, such as the placenta, may be influenced by environmental factors. Placenta can be considered as an intermediate matrix having the potential to express, in association with prenatal exposure to air pollution, distinct biological signatures which may be useful as early-life markers of disease development later in life [49]. Indeed, several studies showed that during pregnancy the exposure to an increased level of PM (PM2.5 or PM10) was associated with changes in placental epigenetic markers (mainly DNA methylation), transcriptomic, and proteomic biomarkers which should be extensively studied to understand the possible consequences on fetus health and adult life [49].

  1. Finally, the authors should put these studies focusing on ASD into the larger context of air pollution and learning disabilities as well as cognitive decline in the elderly.
  2. As suggested, we described the problem of air pollution in a larger context at lines 683-698: “Evidence on the association between air pollution and neurodevelopmental disorders dramatically increased in the last years. Many studies investigated the risk of developing, in addition to ASD, other disorders such as learning disabilities, attention deficit and hyperactivity disorder (ADHD) when children (or their mother) are exposed to high level of airborne pollutants, showing a possible link [45]. The long-term exposure to air pollution begin in the prenatal period and continue during the lifetime, causing the possible onset of chronic adverse outcome such as cancer, cardiovascular and respiratory diseases [46]) Recently, also the cognitive decline in the elderly was associated with atmospheric particles and chemicals [47]. In particular, older adults living in polluted areas showed a decrease in cognitive function and may be at greater risk of experiencing progressive neurodegenerative pathologies such as Alzheimer’s disease [48].

Reviewer 2 Report

The manuscript presents a literature review of studies on early life exposure to air pollution and autism spectrum disorder in children. The review is well presented and I only have some minor comments/suggestions:

  • Figure 1 is missing arrows or lines.
  • It would be helpful to add the geographic region of the study to Table 1-5 and to add the range of the exposure for each study. This could indicate whether there is a threshold of effect.
  • The authors should clarify why they did not carry out a meta-analysis.
  • In the discussion section, it would be useful if the authors could comment on any strengths or weaknesses in individual studies or other factors that may explain the heterogeneity in findings.
  • In some areas of the manuscript (particularly in sections 3 and 4) the English is difficult to understand and may be misleading. For example, the sentence “This association is clearly marked in the 9 case-control studies recruiting a significant sample consisting of 25,631 ASD cases and 155,532 controls, which showed a positive association between PM 2.5  exposure during pregnancy or early life and ASD [23-31].” suggests that the 9 case-control studies were related and the analysis was combined, when they were actually independent studies and there was no combined analysis.

Author Response

Dear Reviewer 2,

thank you so much for your valuable comments and criticisms on our manuscript, which are helpful for revising and improving our paper. The paper titled as "Early-life exposure to environmental air pollution and Autism Spectrum Disorder: a review of available evidence " (ijerph-1023518), by Giovanni Imbriani, Alessandra Panico, Tiziana Grassi, Adele Idolo, Francesca Serio, Francesco Bagordo, Giovanni De Filippis, Donato De Giorgi, Gianfranco Antonucci, Prisco Piscitelli, Manuela Colangelo, Luigi Peccarisi, Maria Rosaria Tumolo, Roberto De Masi, Alessandro Miani and Antonella De Donno, has been reviewed according to reviewers recommendations using the "Track Changes" function in Microsoft Word.

 The comments and the corrections are also outlined below.

The manuscript presents a literature review of studies on early life exposure to air pollution and autism spectrum disorder in children. The review is well presented and I only have some minor comments/suggestions:

  1. Figure 1 is missing arrows or lines. Figure 1 was improved with new arrows as suggested by the reviewer.

  1. It would be helpful to add the geographic region of the study to Table 1-5 and to add the range of the exposure for each study. This could indicate whether there is a threshold of effect. We have added the geographic region where the studies were performed in all the tables. The time window and the average level of airborne contaminants to which people were exposed in each study was also reported in tables. However, the exposure assessment was conducted with different methods among studies, ranging from the assessment of the average exposure to pollutants in the control groups and in the groups of cases, to the evaluation of the increase by interquartile range of each pollutant to which a subject was exposed. For this reason, it is difficult to determine whether an effect threshold exists.

  1. The authors should clarify why they did not carry out a meta-analysis. In our review we have examined the effect of several pollutants but, considering the single pollutants, there were too many few studies to perform an accurate and valid meta-analysis. For this reason, an increase in the number of researchers would improve the statistical power to identify associations between air pollutants and ASD which should be discussed in a future meta-analysis (lines 661-664).

  1. In the discussion section, it would be useful if the authors could comment on any strengths or weaknesses in individual studies or other factors that may explain the heterogeneity in findings. We have discussed the heterogeneity in findings al lines 727-768: “What emerged from the comparison of the studies we selected in this review was the great heterogeneity of the results, which could in part be due to the different number of children recruited, the various statistical methods used for data analysis, and the different ASD assessment systems among studies. The evidence from the case-control studies appeared to be more solid than that from the cohort studies and this could be explained by analyzing the different methods of assessing exposure to air pollution and confounding factors that represented potential bias in some case-control studies.

Confounding due to lifestyle factors, such as smoking in pregnancy, including active smoking and passive smoking, could be problematic, as this factor might be directly associated with an increased risk of the onset of ASD, as reported in several studies [50, 51]. Another potential confounding factor could be that related to socio-economic position at the individual and residential area level. In this context, Miranda et al. found that poorer and minority neighborhoods were more likely to experience higher levels of pollution and were also highly correlated with an increased burden of disease. Despite the important need to adjust to these confounding factors, not all of the studies we reviewed corrected or examined the modification of effects based on the socioeconomic position of individuals or their communities [52].

Since it is necessary to further establish the direct associations between air pollution and adverse neurological effects, such as the onset of ASD, the assessment of exposure to air pollution is a crucial element for future studies. Many studies used a binary exposure measurement, i.e. they compared a high-pollution area versus a low-pollution area by measuring the annual concentrations of air pollutants from the surrounding monitoring stations or taking into account distance from traffic routes as an estimate of atmospheric pollution. These exposure measurements can lead to incorrect exposure classification and, for this reason, it is crucial to develop more accurate methods for measuring chronic exposure to air pollution in this field.

To get a more precise estimate of air pollution exposure, the most accurate method would be to evaluate and quantify individual exposures. Personal monitoring could provide an exposure estimate that is less prone to misclassification than other methods of measuring exposure to air pollution, especially when taking into account well-defined time windows such as the first 1000 days of the life of the child, including nine months of gestation and the first 2 years of the child's life. However, this is more difficult to do if we examine a large cohort of subjects, so this motivation could explain the lower evidence found in the cohort studies.

In this context, a LUR model, can integrate personal monitoring and biomonitoring methods including GIS parameters, such as traffic density, population density, which are used to predict small-scale spatial variation of pollutants [53, 54].  However, this approach has limitations regarding the estimation of air pollution levels referred to the address of residence as ignores historical exposures since people are constantly on the move and are not confined to the place where the exposure was assessed bringing to a potential misclassification of exposure.

In order to reduce the potential for misclassification of exposure, efforts should be made to include information on individual residency history and length of stay in LUR templates.”

  1. In some areas of the manuscript (particularly in sections 3 and 4) the English is difficult to understand and may be misleading. For example, the sentence “This association is clearly marked in the 9 case-control studies recruiting a significant sample consisting of 25,631 ASD cases and 155,532 controls, which showed a positive association between PM 2.5 exposure during pregnancy or early life and ASD [23-31].” suggests that the 9 case-control studies were related and the analysis was combined, when they were actually independent studies and there was no combined analysis. We have revised the English throughout the text and we have rephrased many statements in order to make the manuscript clearer.

Round 2

Reviewer 1 Report

The manuscript has improved and addressed most of my concerns.

This manuscript is an informative summary of the literature. I just wish the authors would have offered a more critical review of the literature. 

Author Response

Dear Reviewer 1,

Thank you for your valuable comments and criticisms on our manuscript, which are helpful for revising and improving our paper. The comments and the corrections are also outlined below.

  1. This manuscript is an informative summary of the literature. I just wish the authors would have offered a more critical review of the literature.
  2. The critical assessment we made of the literature considered in our review has examined various aspects including the comparison between pollutant detection models, proposing a more personalized method (for example through the use of biomarkers) to assess exposure to environmental pollution; we stressed the importance of considering many variables related to lifestyle; we pointed out that different statistical methods were used to highlight associations between pollutants and the development of autism; we have indicated the type of future studies to be undertaken to study this eventual relationship; and we have indicated that at present a causal association between the two phenomena has not been clearly identified. Furthermore, the critical assessment has been improved with another aspect at lines 774-783: “In general, the studies we included in our review had critical points such as the recruitment of small cohorts, the poor comparisons among different areas characterized by different kinds of pollution, the lack of use of a standardized statistical method, the use of different models for assessing the level of exposure to pollutants, and the difficulty of quantify the exposure both for single molecules or their mixture. Furthermore, air pollution is a complex mixture of toxins for which the biological effects on the development of ASD of individual agents are hard to identify considering also the synergistic effects that the various pollutants could have. Therefore, next studies must take into consideration samples to be compared to many areas, ensuring that the participants were exposed to a wide range of pollutant concentrations with individual-level exposure measurements to multiple compounds by various pathways (air, water, diet), combined with genetic information."